# Diastole/Body Mass Index Ratio Can Predict Post-Thoracoscopic Surgery Metastasis in Stage I Lung Adenocarcinoma

**DOI:** 10.3390/jpm13030497

**Published:** 2023-03-09

**Authors:** Hung-Hsing Chiang, Po-Chih Chang, Ting-Wei Chang, Kai-Hua Chen, Yu-Wei Liu, Hsien-Pin Li, Shah-Hwa Chou, Yu-Tang Chang

**Affiliations:** 1Department of Surgery, Division of Chest Surgery, Kaohsiung Medical University Hospital, Kaohsiung 80756, Taiwan; 2Ph.D. Program in Environmental and Occupational Medicine, College of Medicine, Kaohsiung Medical University, Kaohsiung 80756, Taiwan; 3Department of Surgery, College of Medicine, Kaohsiung Medical University, Kaohsiung 80756, Taiwan; 4Department of Surgery, Division of Pediatric Surgery, Kaohsiung Medical University Hospital, Kaohsiung 80756, Taiwan

**Keywords:** body mass index, echocardiogram, lung adenocarcinoma, postoperative metastasis

## Abstract

Background: According to recent animal models for lung adenocarcinoma metastasis, cardiac function may be related to the clinical outcome. The aim of this study is to identify a predictable index for postoperative metastasis (POM) that is associated with cardiac function. Methods. Two hundred and seven consecutive patients who underwent thoracoscopic resection for stage I lung adenocarcinoma were included. Disease-free survival (DFS), overall survival (OS), and patients’ clinical and pathological characteristics were analyzed. Results. Among the 207 patients, 17 cases demonstrated metastasis, 110 cases received a preoperative echocardiogram, and six cases had POM. Mitral valve peak A velocity, which is one of the left ventricular diastolic function parameters affected by BMI (MVPABMI), was associated with a negative factor for POM (hazard ratio (HR): 2.139, *p* = 0.019) and a poor 5-year DFS in the above median (100% vs. 87%, *p* = 0.014). The predictable rate increased from 30.7% to 75% when the MVPABMI was above the median = 3.15 in the solid subtype). Conclusions. MVPABMI is a novel index for POM prediction in early-stage lung adenocarcinoma. This is a pilot study and the first attempt at research to verify that the diastole and the BMI may be associated with POM in early-stage lung adenocarcinoma.

## 1. Introduction

Lung cancer is one of the most prevalent malignant diseases worldwide. In Taiwan, adenocarcinoma is the most common type of cancer, which can occur in several parts of the body [1]. The majority of cases diagnosed are late-stage and have a poor prognosis. However, early-stage patients are usually associated with a better prognosis after radical surgical excision, and their survival rate (SR) reaches 93% [2]. However, regardless of the increased SR, early-stage patients are still at risk for postoperative metastasis (POM), which results in a poorer prognosis than in non-metastatic patients [2].

The echocardiogram is a common imaging screening test for cardiovascular diseases [3] and a popular tool in preoperative cardiac assessments. Among the many advantages of this imaging modality, echocardiograms can provide both structural and functional results, and they are safe, non-invasive, and cost effective. In patients undergoing non-cardiac surgery, echocardiograms can provide more precise information compared to data from their physical examination and history alone. Consequently, based on the information acquired from echocardiograms, surgeons and anesthesiologists can obtain a more detailed examination before the surgical intervention [4]. Although echocardiograms are very useful in terms of assessing the functional status of patients with cancer, they do not have the ability to determine the prognosis of malignant disease. In animal studies with lung cancer distant metastases [5,6], cancer cells must be delivered to the target organs, including the lungs or bones, through the pumping of the heart. Hence, the heart might be an organ that can influence the result of metastasis.

Body mass index (BMI) is a common and useful parameter for population-based studies, including cardiovascular accidents, diabetes mellitus, and other metabolic related diseases [7]. Increased BMI might be related to increased all-cause mortality [8]. In fact, a greater BMI increases the incidence and risk of cancer-related mortality [9,10]. However, in lung cancer, patients with a normal to obese BMI have a better overall survival (OS) rate than underweight patients in stage I non-small cell lung cancer (NSCLC) [7,10,11]. The definitive reasons remain indeterminate, but this is probably related to improved tolerance for further treatment, such as chemotherapy. Nonetheless, there is a distinct lack of data pertaining to disease-free survival (DFS). Hence, our investigation is focused on the relationship between the risk of POM and BMI.

Metastasis after radical resection is a serious condition for early-stage lung cancer patients. Systemic therapy is recommended, and the prognosis is poorer compared to non-metastatic conditions that do not require adjuvant therapy [2]. Parameters for NSCLC surveillance, including preoperative biomarkers (serum carcinoembryonic antigen (CEA), serum cytokeratin 19 fragment (CYFRA 21-1)), tumor size, cell subtypes, and differentiation, are useful prognostic markers [12,13]. Currently, regular surveillance with imaging modalities and/or serum biomarkers without any adjuvant therapy is recommended after radical excision in patients with early-stage lung cancer. Current literature supports that although adjuvant therapy can improve survival rates in high-risk cases, this approach may also increase the risk of harmful side effects in patients [14]. Hence, more useful, predictable parameters are needed for precision and prevention of metastasis in patients with a high risk of metastasis.

Consequently, we are curious about the relationship between heart function and BMI for POM in early-stage lung adenocarcinoma. This study attempted to identify high risk POM from cardiac function and BMI.

## 2. Materials and Methods

### 2.1. Study Population

From January 2014 to December 2018, lung adenocarcinoma patients who underwent curative surgery were investigated at the division of thoracic surgery, department of surgery, Kaohsiung medical university hospital (KMUH), Kaohsiung, Taiwan. If there were any recommendations or necessary preoperative assessments, an echocardiogram was performed by qualified cardiologists before general anesthesia. The left heart function was the main evaluation target. A complete staging workup was performed on all patients, including chest computed tomography (CT), brain magnetic resonance imaging (MRI), and a whole-body bone scan for operation, planning, and evaluation of distant metastases. TNM stage and POM were classified according to the American Joint Committee of Cancer Eighth Edition (AJCC 8th) Lung Cancer Staging system [15].

All patients underwent minimally invasive thoracoscopic surgery under double lumen general anesthesia, with one lung ventilated in the decubitus position. Surgical procedures, including wedge resection, segmentectomy (sublobar group), lobectomy, and bilobectomy (lobectomy group), had been previously performed after complete evaluation and assessment by each surgeon in KMUH. No patient underwent a pneumonectomy. The section margins were all free from the lesion. Systemic mediastinal lymph node sampling or dissection was also performed.

Histologic subtypes of lung adenocarcinoma were classified in line with the new International Association for the Study of Lung Cancer, American Thoracic Society, and European Respiratory Society (IASLC/ATS/ERS) multidisciplinary lung adenocarcinoma classification [16]. Lymphovascular invasion was defined as the identification of tumor cells in the lymphatic or blood vessel lumen. Histologic grade and spread through air space (STAS) were determined according to the 2015 World Health Organization (WHO) classification [16]. Pathologic classifications were conducted independently by two qualified pathologists, and rare discrepancies were resolved through reexamination of the slides and discussion.

Patients’ clinical characteristics, including gender, age, BMI, history of cardiovascular (CV) diseases, and blood pressure (BP), were collected. Pathological stages IA to IB had been included, and the cancer staging was the AJCC 8th edition. Patients who received any kind of adjuvant therapy or neoadjuvant therapy and had a history of malignant tumors were excluded. Finally, 207 patients were included in this study.

### 2.2. Patient Follow-Up Evaluation

The surveillance period was until December 2020, and the median follow-up period was 57 months. The surveillance schedule was as follows: follow-up in the clinic every three to four months for the first two years after the operation, every six months for three to five years, and every 12 months thereafter. A chest CT was routinely arranged at each surveillance visit for inspecting potential local recurrence and lung metastasis. In non-symptomatic patients, brain MRI, liver sonography, and bone scans were arranged every six to 12 months for the detection of distant metastasis. If any symptoms were suspicious during the surveillance, the above examinations were immediately arranged to exclude POM. DFS was defined as the time from the date of surgery to a recurrence or metastasis of the disease. OS was defined as the time from the date of surgery until death. Patients without recurrence were censored at the time of their last negative follow-up or their death without evidence of recurrence. Study inclusion criteria were pathologically confirmed stage IA/B (T1a/T1b/T1c/T2aN0M0) adenocarcinoma and complete resection with systemic lymph node sampling or dissection. This study was performed in accordance with the Helsinki Declaration. Our institutional review board approved this study (approval number: KMUHIRB-E(II)-20220075). Informed consent was waived because this was a retrospective study.

### 2.3. Statistics

We used the IBM SPSS 19.0 edition software to perform all statistical analyses. All tests, including independent t and two-tailed Chi tests, were analyzed for the identification of any potential differences. The echocardiogram results were adjusted according to BMI. After the identification of differences, we used the cox regression model for prognostic factor identification, including univariate and multivariate analysis. The Kaplan–Meier method was used for the analysis of the 5-year DFS and OS, and log-rank tests were used for comparisons of DFS and OS between two categories in univariate analysis. Variables with a *p* value of less than 0.05 were entered into the survival analysis and cox regression.

## 3. Results

A total of 273 patients conformed to the study’s eligibility criteria. Sixty patients who received adjuvant therapy, four patients who were categorized as adenocarcinomas in situ, and two patients who received neoadjuvant therapy were excluded. Consequently, 207 patients were included in the study, of whom 115 were female and 92 were male (Figure 1). 

The definition of POM was classified according to the AJCC’s 8th edition M status after surgery. Finally, 11 female and six male patients had POM. The lung was the most popular metastatic site, whereas metastasis in the brain, pleura, adrenal gland, bone, and liver was also observed. The mean DFS time was 23.59 months (6–53 months) in the POM group. The median ages of the non-metastatic group and POM were 60.36 and 62.41 years, respectively, at the time of diagnosis. There was no POM in the T1a and lepidic subtypes. According to the literature, BMI is a factor for lung cancer prognosis, and a high BMI has a better outcome compared to a low BMI. However, there are no definitive data for POM. To investigate the relationship between BMI and POM, we collected patients’ BMI data, and our findings showed that patients in the metastatic and non-metastatic groups were 23.54 and 24.04, respectively. The above data showed non-significant differences (Table 1).

Surgical types, including wedge resection, segmentectomy (sublobar group), lobectomy, and bilobectomy (lobectomy group), are not a prognostic factor for POM. In the data from surgical specimens’, we were missing two cases of adenocarcinoma subtypes. Adenocarcinoma subtypes, including papillary and solid, visceral pleural invasion (T2a), and tumor size were independent factors for increased probability of POM (HR: 4.089, 4.070, 3.485, and 3.636, respectively). The percentage of each subtype for each lesion revealed that papillary and solid components increased the risk of POM (HR: 1.019 and 1.031, respectively), whereas the lepidic subtype percentage reduced the risk of POM (HR: 0.96) (Table 2).

The Kaplan–Meier survival curve also revealed differences on the 5-year DFS in T2a, papillary, and solid subtypes (85%, 75%, and 72%, respectively, *p* < 0.05) but not in OS (Figure 2).

Kaplan–Meier survival curves of the solid subtype, T2a, and papillary subtype. Notice that each parameter increases the incidence of postoperative metastasis (68%, 85%, and 75%, respectively, *p* < 0.05). No significant difference, except for the solid subtype, was identified in overall survival (80%, *p* < 0.05).

Preoperative echocardiogram data were collected and analyzed to evaluate whether cardiac function influences POM in early-stage lung adenocarcinoma. We performed 207 echocardiograms in 110 patients, and our findings showed that no patients demonstrated severe cardiac function abnormalities, including valvular diseases. Furthermore, there were no significant differences, including cardiovascular diseases, between the echocardiogram group and the non-echocardiogram group (Table 3).

Among the 110 patients, six patients had POM, with significant differences in the percentages of lepidic and solid subtypes (Table 4).

The underlying target of the echocardiogram was to perform a preoperative assessment, and thus left cardiac function was the main evaluation target. No patients had moderate-to-severe valvular diseases or left ventricular systolic function impairment. We analyzed all the results from the echocardiogram, and the data revealed that there were no significant differences in any of the investigated parameters between the two groups. Because BMI influences lung cancer survival [11], we adjusted the echocardiogram data according to the BMI. Consequently, we noticed significant differences between POM and non-POM groups when the mitral valve peak A and E velocities were adjusted by BMI (MVPABMI and MVPEBMI) (Table 5).

MVPABMI and MVPEBMI were independent factors for increased POM (HR: 2.139 and 2.293, respectively), and only MVPABMI was associated with the solid subtype (Table 6).

We noticed that all POMs occurred up to a median level of 3.15. The predictable rate increased from 30.7% to 75% when the MVPABMI reached 3.15 in the solid subtype. Furthermore, only the DFS of MVPABMI (87%) showed significant differences in survival but not in OS (Figure 3).

Notice that the Kaplan–Meier survival curve of the diastole/body mass index ratio (MVPABMI) up to 3.15 increased the incidence of postoperative metastasis (100% vs. 87%, *p* = 0.014).

Since CV diseases and BP might influence the left ventricular diastolic function, we investigated the above parameters, and our findings revealed no significant differences (Table 2 and Table 3). Thus, left ventricular diastolic function and BMI might influence POM in resected early lung adenocarcinoma, especially in the solid subtype.

## 4. Discussion

In this study, we found that MVPABMI can predict POM in stage I lung adenocarcinoma before resection, especially in the solid subtype. There were no participants with metastasis or recurrence in tumor size less than 1 cm in diameter without visceral pleural invasion (T1a). In fact, these two parameters can be easily collected before any pathological diagnosis. In parameters from surgical specimens, the solid, papillary, and micropapillary predominant subtypes increase the metastatic incidence, i.e., metastasis does not occur in the lepidic subtype. In contrast to the lepidic components, the percentage of solid components in pathological examination increases the incidence of metastasis. Surgical types, including lobectomy and sublobar excision, were not found to influence POM in our cohort. Taken together, the solid subtype, tumor size, visceral pleural invasion, and MVEAPBMI were found to be independent factors for POM in early-stage lung adenocarcinoma. MVPABMI is related to the solid subtype but not to visceral pleural invasion. This in turn suggests that the MVPABMI value increases the probability of metastasis in the solid subtype.

According to the WHO classification for lung tumors, the subtype of lung adenocarcinoma can predict the subsequent treatment outcome. However, the solid subtype has been found to be an independent negative predictive factor for outcomes, including OS and post-recurrent survival [16]. In our study, all cases were stage I, the tumor size was below 3 cm in diameter, and the solid subtype was found to be an independent negative factor. DFS and OS were shown to be poorer than other subtypes. As shown in previous studies, pleural invasion and tumor size are also independent factors that indicate a poorer outcome [17,18]. Our results are comparable with the findings of these research studies in the solid subtype, visceral pleural invasion, and tumor size.

BMI is widely used in obesity studies. Obesity is linked to metabolic diseases, including cardiovascular events, diabetes mellitus, hypertension, and hyperlipidemia. Greater BMI is also related to higher morbidity and mortality rates [8]. In breast cancer, increased BMI results in a poor prognosis as a result of a complex mechanism involving inflammation, hormone, and adipokines [9,19]. In contrast, normal to overweight patients have better lung cancer outcomes compared to underweight patients. This may be related to patients’ tolerance for anti-cancer therapy, including anti-cancer agents, radiotherapy, and surgical interventions [11]. In our study, we focused on POM in stage I lung adenocarcinoma, and we did not identify any relationship between BMI and metastasis. After adjusting mitral valve peak A velocity data by BMI, our data revealed a significant difference between patients with and without metastasis. Thus, we support the idea that BMI might influence POM.

An echocardiogram is an easy-to-use, non-invasive tool for cardiac function assessments. In KMUH, echocardiograms are recommended before operations on patients with a high-risk presentation of CV diseases in accordance with their medical histories. If any abnormalities are detected by the echocardiogram, further examinations, including a single photon emission CT, a treadmill test, or coronary artery catheterization, are scheduled by the cardiologist. In our study, no patients underwent additional examinations for the heart because echocardiography revealed a normal, limited left ventricular systolic function.

In animal studies, intracardiac lung cancer cell line injection to the left ventricle can induce bone or brain metastasis, whereas pulmonary metastasis can be induced by tail vein injection [5,6]. The above models revealed that the heart may be a prominent factor in metastasis. Thus, we support the notion that cardiac function is related to lung cancer metastasis. In our cases, the left cardiac diastolic function might indirectly influence metastasis in lung adenocarcinoma, as evidenced by the mitral valve peak E and A velocities from the preoperative echocardiography data after adjustment for BMI. Mitral valve peak E and A velocities reflect left ventricular diastolic function. These parameters represent the pressure gradient in early and late diastole of the left heart, respectively. [20]. In patients with myocardial disease, LV diastolic dysfunction can be predicted by changing the E:A ratio through LV filling pressure [21]. In our study, no patient had a myocardial event before resection. These findings may be explained by the increased pulmonary venous flow velocity, which may facilitate the translocation of cancer cells to the pulmonary vein.

MVPABMI, which combines BMI and diastolic function, is a novel index for POM prediction in early-stage adenocarcinoma. We support the idea that MVPABMI is dictated by a complex mechanism and that multiple factors facilitate POM. Recently, many studies have concluded that air pollution is a risk factor for lung cancer [22,23]. Moreover, air pollution is also related to the development of cardiovascular disease and diastolic and left ventricular dysfunction because pollutants, including fine particulate matter, polycyclic aromatic hydrocarbons, and NOx, induce oxidative stress and chronic inflammation [22,23]. In Taiwan, poorer outcomes for lung cancer have been shown in industrial cities compared to non-industrial ones [1]. Inflammation has also been shown to influence diastolic dysfunction in an animal study [24]. Altogether, air pollution is probably one of the main causes of POM as a result of diastolic dysfunction in lung adenocarcinoma. However, further prospective or animal studies are necessary to confirm this speculation.

Our study has several limitations. First, our study is retrospective and has a relatively small sample size. Only six patients who received a preoperative echocardiogram had POM. Pathologic classifications, including histological grade, lymphovascular invasion, and STAS, were established as independent prognostic factors [25,26,27]. Before 2016, the above data were not routinely evaluated in KMUH, which unavoidably resulted in incomplete evaluations and thus were not related to MVPABMI. Secondly, our study did not include any information on other well-established independent factors for POM, such as tumor markers and genetic abnormalities. Therefore, we do believe that a multicenter study is necessary for increasing the number of cases included and designing a prospective study to further validate our findings.

## 5. Conclusions

In conclusion, this study provides proof that both diastolic function and BMI influence lung adenocarcinoma POM. Our results suggest that lung adenocarcinoma is associated with cardiac function and BMI, which in turn act as triggers of POM.

## Figures and Tables

**Figure 1 jpm-13-00497-f001:**
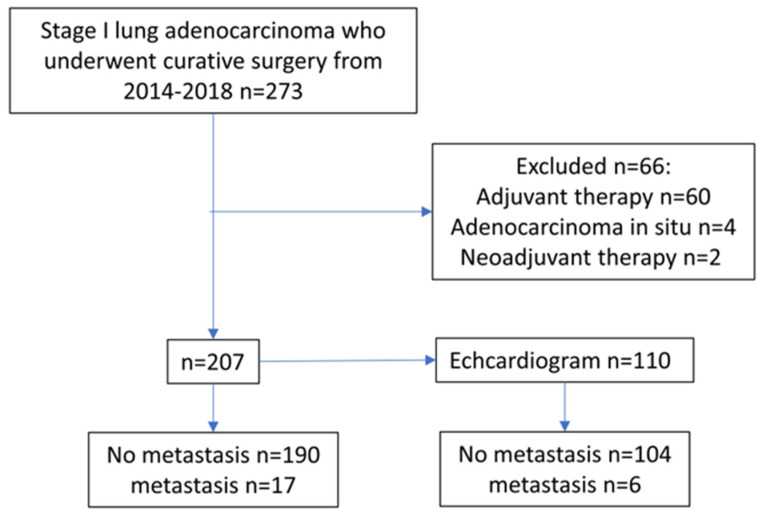
Flow chart of the study.

**Figure 2 jpm-13-00497-f002:**
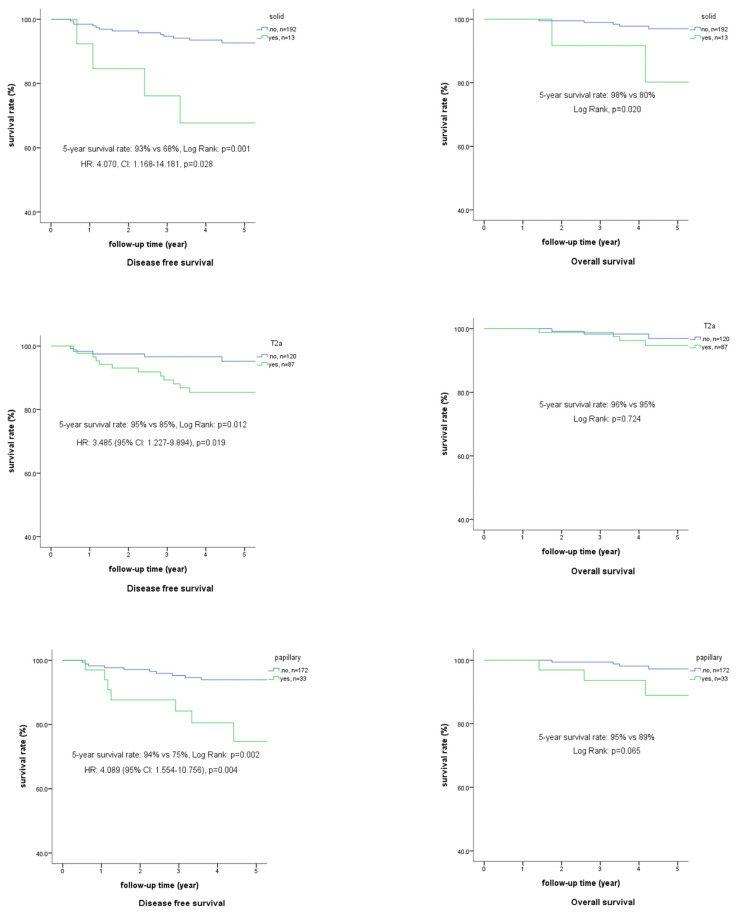
Survival curves for each of the independent factors for postoperative metastasis.

**Figure 3 jpm-13-00497-f003:**
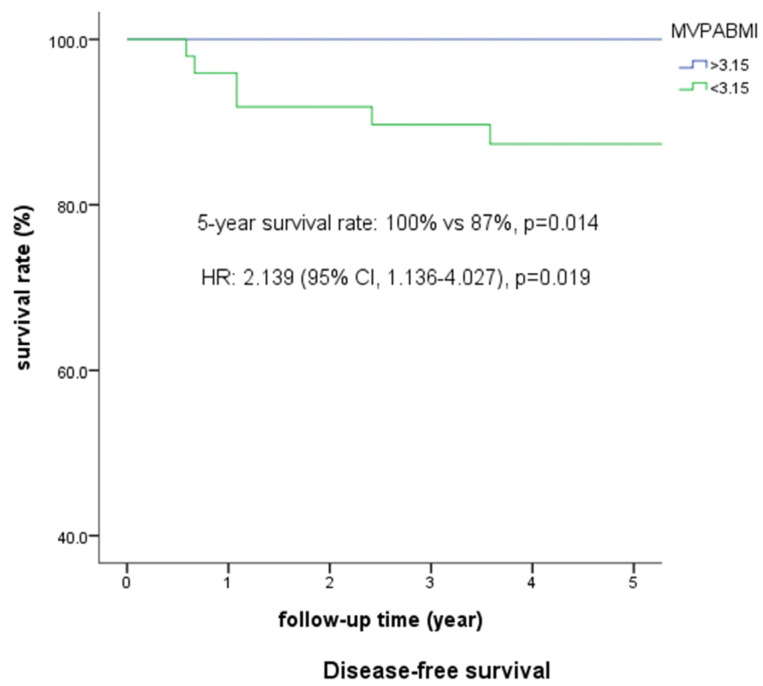
Disease-free survival of the diastole/body mass index ratio.

**Table 1 jpm-13-00497-t001:** Comparison between non-metastatic and POM groups.

		No Metastasis (N = 190)	POM (N = 17)	*p-*Value
SEX (%)	Male (44.4)	86 (93.5)	6 (6.5)	0.43
	Female (55.6)	104 (90.4)	11 (9.6)	
AGE (YEAR-OLD)		60.36 ± 11.19	62.41 ± 8.04	0.46
BMI (KG/M^2^)		24.04 ± 3.99	23.54 ± 2.99	0.61
SURGICAL TYPE (%)	Lobectomy (60.7)	115 (91.3)	11 (8.7)	0.74
	Sublobar (39.3)	75 (92.6)	6 (7.4)	
TUMOR SIZE (CM)		1.52 ± 0.64	2.04 ± 0.56	0.001
ADENOCARCINOMA SUBTYPE (%)	Lepidic (21.7)	45 (100)	0 (0)	<0.001
	Acinar (54.6)	106 (94.6)	6 (5.4)	0.050
	Papillary (15.9)	26 (78.8)	7 (21.2)	0.022
	Micropapillary (1.4)	2 (66.7)	1 (33.3)	0.45
	Solid (6.4)	9 (75)	3 (25)	0.027
T STATUS (%)	T1a (31.9)	66 (100)	0 (0)	0.003
	T1b (21.7)	42 (93.3)	3 (6.7)	0.67
	T1c (4.3)	7 (77.8)	2 (22.2)	0.19
	T2a (42.1)	75 (86.2)	12 (13.8)	0.013
TUMOR GRADE (%)	Well (10.1)	21 (100)	0 (0)	0.15
	Moderate to poor (89.9)	169 (90.9)	17 (9.1)	
STAS (%)	Not identified (35.7)	72 (97.3)	2 (2.7)	0.070
	Present (13.0)	25 (92.6)	2 (7.4)	
	Not mentioned (51.3)	93	13	
LYMPHOVASCULAR INVASION (%)	Not identified (89.7)	172 (93.0)	13 (7.0)	0.17
	Present (8.7)	15 (83.3)	3 (16.7)	
	Not mentioned (1.6)	3	1	
PERCENTAGE OF ADENOCARCINOMA SUBTYPE	Lepidic	27.89 ± 29.01	10.00 ± 12.87	<0.001
	Acinar	49.05 ± 29.34	34.41 ± 28.88	0.050
	Papillary	14.87 ± 22.16	27.94 ± 25.25	0.022
	Micropapillary	1.95 ± 8.49	4.71 ± 14.63	0.45
	Solid	5.05 ± 14.72	22.35 ± 29.05	0.027
BLOOD PRESSURE (MMHG)	Systolic	134.82 ± 18.70	133.77 ± 15.12	0.85
	Diastolic	78.39 ± 11.49	77.85 ± 13.17	0.87
CARDIOVASCULAR DISEASES (%)	Yes (33.8)	65 (92.9)	5 (7.1)	0.69
	No (66.2)	125 (91.2)	12 (8.8)	

BMI = body mass index, POM = post-operative metastasis, STAS = spread through air space.

**Table 2 jpm-13-00497-t002:** Regression of each difference between the non-metastatic and POM groups.

Univariate Regression Analysis				
	HR	95% CI		*p-*Value
TUMOR SIZE	3.636	1.595	8.290	0.002
T1A	0.026	0.00	1.784	0.091
T2A	3.485	1.227	9.894	0.019
SUBTYPE				
LEPIDIC	0.033	0.00	3.884	0.16
ACINAR	0.436	0.161	1.180	0.10
PAPILLARY	4.089	1.554	10.756	0.004
SOLID	4.070	1.168	14.181	0.028
PERCENTAGE				
LEPIDIC	0.960	0.928	0.994	0.020
ACINAR	0.983	0.966	1.000	0.053
PAPILLARY	1.019	1.003	1.036	0.022
SOLID	1.031	1.016	1.047	<0.001

CI = confidence interval, HR = hazard ratio.

**Table 3 jpm-13-00497-t003:** Comparison between the echocardiogram and non-echocardiogram groups.

		Non-Cardioechogram (N = 97)	Cardioechogam (N = 110)	*p-*Value
SEX (%)	Male (44.4)	38 (41.3)	54 (58.7)	0.15
	Female (55.6)	59 (51.3)	56 (48.7)	
AGE (YEAR-OLD)		60.32 ± 11.18	60.71 ± 10.82	0.80
BMI (KG/M^2^)		24.20 ± 3.38	23.83 ± 4.34	0.50
SURGICAL TYPE (%)	Lobectomy (60.9)	64 (50.8)	62 (49.2)	0.16
	Sublobar (39.1)	33 (40.7)	48 (59.3)	
TUMOR SIZE (CM)		1.58 ± 0.66	1.54 ± 0.63	0.61
ADENOCARCINOMA SUBTYPE (%)	Lepidic (21.7)	23(51.1)	22 (48.9)	0.52
	Acinar (54.6)	53 (46.9)	60 (53.1)	0.98
	Papillary (15.9)	16 (48.5)	17 (51.5)	0.84
	Micropapillary (1.4)	3 (75)	1 (25)	0.25
	Solid (6.4)	4 (30.8)	9 (69.2)	0.23
PERCENTAGE OF ADENOCARCINOMA SUBTYPE (%)	Lepidic	28.09 ± 29.60	24.95 ± 27.45	0.43
	Acinar	46.96 ± 30.14	48.64 ± 29.05	0.68
	Papillary	15.52 ± 23.77	16.32 ± 21.71	0.80
	Micropapillary	3.20 ± 12.53	1.27 ± 4.20	0.15
	Solid	5.10 ± 14.12	7.68 ± 19.07	0.27
T STATUS (%)	T1a (31.9)	37 (56.1)	29 (43.9)	0.070
	T1b (21.7)	16 (35.6)	29 (64.4)	0.086
	T1c (4.3)	2 (22.2)	7 (77.8)	0.13
	T2a (42.1)	42 (48.3)	45 (51.7)	0.73
METASTASIS (%)	No (91.8)	86 (45.3)	104 (54.7)	0.12
	Yes (8.2)	11 (64.7)	6 (35.3)	
BLOOD PRESSURE (MMHG)	Systolic	134.15 ± 17.85	135.27 ± 18.94	0.72
	Diastolic	77.68 ± 11.79	78.99 ± 11.85	0.50
CARDIOVASCULAR DISEASES (%)	Yes (33.8)	30 (42.9)	40 (57.1)	0.41
	No (66.2)	67 (48.9)	70 (51.1)	

BMI = body mass index.

**Table 4 jpm-13-00497-t004:** Comparison of metastasis and POM in the echocardiogram group.

		No Metastasis (n = 104)	POM (N = 6)	*p-*Value
SEX (%)	Male (51.9)	51 (94.4)	3 (5.6)	0.96
	Female (48.1)	53 (94.6)	3 (5.4)	
AGE (YEAR-OLD)		60.79 ± 11.04	59.33 ± 6.44	0.75
BMI (KG/M^2^)		24.23 ± 3.77	21.45 ± 2.89	0.078
SURGICAL TYPE (%)	Lobectomy (54.8)	57 (91.9)	5 (8.1)	0.17
	Sublobar (45.2)	47 (97.9)	1 (2.1)	
TUMOR SIZE (CM)		1.51 ± 0.63	2.05 ± 0.50	0.042
ADENOCARCINOMA SUBTYPE (%)	Lepidic (20)	22 (100)	0 (0)	0.21
	Acinar (54.6)	59 (98.3)	1 (1.7)	0.052
	Papillary (15.4)	15 (88.2)	2 (11.8)	0.22
	Micropapillary (0.9)	1 (100)	0 (0)	0.81
	Solid (8.1)	6 (66.7)	3 (33.3)	<0.001
PERCENTAGE OF ADENOCARCINOMA SUBTYPE (%)	Lepidic	25.96 ± 27.78	7.5 ± 11.73	0.11
	Acinar	49.81 ± 28.81	28.33 ± 27.87	0.078
	Papillary	15.82 ± 21.44	25.00 ± 26.65	0.32
	Micropapillary	1.35 ± 4.31	0.00 ± 0.00	0.45
	Solid	5.87 ± 15.43	39.17 ± 42.00	<0.001
T STATUS (%)	T1a (26.4)	29 (100)	0 (0)	0.13
	T1b (26.4))	27 (93.1)	2 (6.9)	0.69
	T1c (6.3)	7 (100)	0 (0)	0.51
	T2a (40.9)	41 (91.1)	4 (8.9)	0.19

BMI = body mass index, POM = post-operative metastasis.

**Table 5 jpm-13-00497-t005:** Comparison of echocardiogram data between the nonmetastatic and the POM groups.

	NO (N=)	POM (N=)	*p-*Value
Aortic opening diameter (cm)	3.097 ± 0.461	3.010 ± 0.288	0.65
	(102)	(6)	
Left atrial diameter (cm)	3.442 ± 0.765	3.352 ± 0.316	0.78
	(102)	(6)	
Aortic valve cusp opening (cm)	1.813 ± 0.292	1.860 ± 0.232	0.65
	(100)	(6)	
Left atrium to aortic root ratio	1.129 ± 0.279	1.118 ± 0.117	0.92
	(102)	(6)	
Interventricular septal end diastole (cm)	0.980 ± 0.226	0.948 ± 0.229	0.75
	(104)	(5)	
Left ventricular internal diameter end diastole (cm)	4.699 ± 0.679	4.646 ± 0.971	0.87
	(104)	(5)	
Left ventricular posterior wall end diastole (cm)	0.936 ± 0.184	0.928 ± 0.198	0.93
	(104)	(5)	
Interventricular septal end systole (cm)	1.396 ± 0.314	1.418 ± 0.536	0.88
	(104)	(5)	
Left ventricular internal diameter end systole (cm)	2.814 ± 0.581	2.894 ± 0.729	0.77
	(104)	(5)	
Left ventricular posterior wall end systole (cm)	1.458 ± 0.268	1.332 ± 0.361	0.31
	(104)	(6)	
End diastolic volume (mL)	105.418 ± 34.443	101.440 ± 47.644	0.79
	(104)	(6)	
End systolic volume (mL)	31.999 ± 15.515	33.460 ± 18.822	0.83
	(104)	(6)	
Stroke volume (mL)	73.418 ± 23.915	67.980 ± 29.746	0.59
	(104)	(6)	
Ejection fraction (%)	70.414 ± 8.601	68.057 ± 6.185	0.51
	(104)	(6)	
Fraction shortening (%)	40.370 ± 7.249	38.066 ± 5.347	0.49
	(104)	(5)	
Mitral valve peak A velocity (MV Peak A Vel) (cm/s)	77.5 ± 22.7	92.7 ± 20.9	0.11
	(91)	(6)	
Mitral valve peak E velocity (MV Peak E Vel) (cm/s)	69.9 ± 17.2	80.4 ± 17.6	0.15
	(92)	(6)	
MVPABMI (MV Peak A Vel/BMI)	3.243 ± 1.094	4.380 ± 1.090	0.015
	(91)	(6)	
MVPEBMI (MV Peak E Vel/BMI)	2.948 ± 0.901	3.806 ± 0.963	0.027
	(92)	(6)	

BMI = body mass index, MV = mitral valve, POM = post-operative metastasis, Vel = velocity.

**Table 6 jpm-13-00497-t006:** Regression of each difference between the non-metastatic and POM groups in the echocardiogram group.

	Univariate Regression Analysis				Multiple Regression Analysis			
	HR	95% CI		*p*	HR	95% CI		*p*
MVPABMI MVPEBMI	2.139 2.293	1.136 1.043	4.027 5.040	0.019 0.039		(reference)		
SOLID (SUBTYPE)	12.937	2.599	64.387	0.002				
SOLID (PERCENTAGE)	1.043	1.019	1.068	<0.001	1.044	1.019	1.069	<0.001

CI = confidence interval, HR = hazard ratio, MVPABMI = mitral valve peak velocity A/body mass index, MVPEBMI = mitral valve peak velocity E/body mass index.

## Data Availability

The data presented in this study are available on request from the corresponding author.

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
