# Peer review of "Diastole/Body Mass Index Ratio Can Predict Post-Thoracoscopic Surgery Metastasis in Stage I Lung Adenocarcinoma"

_jpm, 2023, doi:10.3390/jpm13030497_

Round 1

Reviewer 1 Report

Chiang et al. have investigated the body mass index (BMI) as a predictive marker for post-thoracoscopic surgery metastasis in stage I lung adenocarcinoma. The rationale for the study is based on animal study suggesting a correlation as well as the rationale that blood has to pass through heart for lung adenocarcinoma metastasis to other organs. Authors retrospectively analyzed echocardiogram data from patients that had metastasis against those that did not had metastasis. Surprisingly, in contrast to the hypothesize and shown association between BMI and lung cancer patient survival, authors did not find any difference between patients with metastasis and non-metastatic patients. However, when the echocardiogram data were adjusted for the patient BMI, authors found a significant difference in the mitral valve peak A and E velocity for post operative metastasis. Overall, the study is well performed, and the data supports their interpretation. The study has limitation but they have been acknowledged by the authors. The study does not have extraordinary finding but the finds are worth sharing with the research community and meet the requirements of scientific inquiry and shall be accepted with minor revision.

Major comments:

1.       What does MVPABMI and MVPEBMI means? What does that suggest? For example, MVPE/MVPA ratio assess diastolic filling. Does MVPABMI and MVPEBMI mean anything?

Minor comment:

1.       In Line 248, ‘Table 5. Comparison of echocardiogram data from the metastatic and the POM group’ should be ‘Table 5. Comparison of echocardiogram data between non-metastatic and the POM group’.

Author Response

Major comment:  What does MVPABMI and MVPEBMI means? What does that suggest? For example, MVPE/MVPA ratio assess diastolic filling. Does MVPABMI and MVPEBMI mean anything?

Response: MVPA and MVPE means left ventricular diastole indexes and higher BMI emerges impaired diastolic filling indexes. MVPABMI and MVPEBMI means patient’s left ventricular diastole status according to two diastole indexes combination (MVPA or MVPE adjusts BMI). Higher MVPABMI or MVPEBMI emerges higher incidence of postoperative metastasis in stage I lung adenocarcinoma.

Minor comment: In Line 248, ‘Table 5. Comparison of echocardiogram data from the metastatic and the POM group’ should be ‘Table 5. Comparison of echocardiogram data between non-metastatic and the POM group’.

Response:  We have rewritten this according to your suggestion.

Reviewer 2 Report

Dear Authors,

thank you for giving me the opportunity of reviewing this interesting paper which introduces a novel parameter for the prediction of lung cancer metastasis. The paper is well-written and the methodology is clearly presented. I did not detect any significant bias in the analysis, limitations are well described and I think that this study represents a nice starting point for future larger investigations, which may reveal some interessting aspects on cancer metastasis pathogenesis. I would therefore accept the paper in its present form. I only suggest correcting the graphs titles in figure 2, as I think the title is wrong in the left-center image.

Thank you again.

Author Response

Point 1: thank you for giving me the opportunity of reviewing this interesting paper which introduces a novel parameter for the prediction of lung cancer metastasis. The paper is well-written and the methodology is clearly presented. I did not detect any significant bias in the analysis, limitations are well described and I think that this study represents a nice starting point for future larger investigations, which may reveal some interessting aspects on cancer metastasis pathogenesis. I would therefore accept the paper in its present form. I only suggest correcting the graphs titles in figure 2, as I think the title is wrong in the left-center image.

Thank you again.

response: Thank you again for your valuable comments on our manuscript. We have made changes and corrections to our article accordingly.
